# Fatty Acid ABCG Transporter *GhSTR1* Mediates Resistance to *Verticillium dahliae* and *Fusarium oxysporum* in Cotton

**DOI:** 10.3390/plants14030465

**Published:** 2025-02-05

**Authors:** Guanfu Cheng, Xiuqing Li, W. G. Dilantha Fernando, Shaheen Bibi, Chunyan Liang, Yanqing Bi, Xiaodong Liu, Yue Li

**Affiliations:** 1Key Laboratory of Biological Ecological Adaptation and Evolution in Extreme Environments, College of Life Science, Xinjiang Agricultural University, Urumqi 830001, China; cgfyouxi@163.com (G.C.); 15099389419@163.com (X.L.); lcyiang060723@163.com (C.L.); b916228271@126.com (Y.B.); xiaodongliu75@aliyun.com (X.L.); 2Department of Plant Science, University of Manitoba, Winnipeg, MB CAR3T2N2, Canada; dilantha.fernando@umanitoba.ca (W.G.D.F.); shaheen.bibi@umanitoba.ca (S.B.)

**Keywords:** *GhSTR1*, ABCG transporter, cotton disease resistance, Verticillium wilt and Fusarium wilt, growth–defense trade-off

## Abstract

Verticillium wilt and Fusarium wilt cause significant losses in cotton (*Gossypium hirsutum*) production and have a significant economic impact. This study determined the functional role of *GhSTR1*, a member of the ABCG subfamily of ATP-binding cassette (ABC) transporters, that mediates cotton defense responses against various plant pathogens. We identified *GhSTR1* as a homolog of *STR1* from *Medicago truncatula* and highlighted its evolutionary conservation and potential role in plant defense mechanisms. Expression profiling revealed that *GhSTR1* displays tissue-specific and spatiotemporal dynamics under stress conditions caused by *Verticillium dahliae* and *Fusarium oxysporum*. Functional validation using virus-induced gene silencing (VIGS) showed that silencing *GhSTR1* improved disease resistance, resulting in milder symptoms, less vascular browning, and reduced fungal growth. Furthermore, the *AtSTR1* loss-of-function mutant in *Arabidopsis thaliana* exhibited similar resistance phenotypes, highlighting the conserved regulatory role of *STR1* in pathogen defense. In addition to its role in disease resistance, the mutation of *AtSTR1* in *Arabidopsis* also enhanced the vegetative and reproductive growth of the plant, including increased root length, rosette leaf number, and plant height without compromising drought tolerance. These findings suggest that *GhSTR1* mediates a trade-off between defense and growth, offering a potential target for optimizing both traits for crop improvement. This study identifies *GhSTR1* as a key regulator of plant–pathogen interactions and growth dynamics, providing a foundation for developing durable strategies to enhance cotton’s resistance and yield under biotic and abiotic stress conditions.

## 1. Introduction

Cotton (*Gossypium hirsutum*) is an economically important crop that plays a key role in the global textile industry [1]. However, vascular diseases, mainly Verticillium wilt and Fusarium wilt, significantly affect its production. These diseases have resulted in 10–35% annual yield losses in cotton production and reduced fiber quality [2]. Verticillium wilt and Fusarium wilt are caused by the soil-borne pathogens *Verticillium dahliae* and *Fusarium oxysporum*, respectively. These two pathogens are significant issues in cotton production in China, especially in Xinjiang province [3]. Both pathogens invade plants through the roots and establish infection in the vascular tissue, thus disrupting water transport and inducing symptoms, such as leaf chlorosis and systemic wilt [2,4]. Cultural practices and fungicide applications have been used to mitigate these diseases. However, these methods are not entirely effective in managing pathogens, and using fungicides often leads to fungicide resistance, high costs, and high environmental risks [5]. Plant breeding, which focuses on genetic resistance, has emerged as a promising alternative to the traditional methods for mitigating these diseases. Genetic resistance is more effective for cotton production and promotes environmentally sustainable agricultural practices [6]. However, candidate resistance resources are limited for genetic resistance in Cotton against *Verticillium dahliae* and *Fusarium oxysporum* [3,7]. Given the substantial economic and agricultural impact of these diseases, it is essential to identify more genes related to cotton defense and use them for cotton breeding and resistance [8].

Plants can prevent pathogen invasion and limit their growth through innate immunity, where the immune system relies on signal transduction pathways [9]. Innate immunity initiates a regulatory mechanism that overcomes pathogen invasion. Plants enhance their resistance to pathogen invasion by fortifying the cell wall, reprogramming metabolic pathways, and activating signal transduction cascades that restrict pathogen colonization and dissemination [10,11]. Pathogens deploy effector proteins and toxins to inhibit host defense mechanisms. This approach enables pathogens to establish infections and use their carbon and energy resources to support pathogen multiplication [12]. Fatty acids, the key energy reserve structural components in plants, serve as a resource for competition between plants and pathogens [13]. During pathogen infection, plants enhance their disease resistance through the stringent use of these resources [14]. However, pathogens often counteract these defenses by activating host fatty acid transporters, enabling them to “hijack” fatty acids directly from plant cells for their growth and infection processes [15]. In addition to their function in plant–pathogen interactions, fatty acids play a significant role in symbiotic relationships. For example, arbuscular mycorrhizal fungi (AMF) depend on their symbiotic association with host plants to obtain fatty acids as their primary carbon source [16,17]. This dual role of fatty acid transport makes them key constituents for plant defense mechanisms against pathogens and as a regulatory element in maintaining plant–microbe symbiosis.

Fatty acid transport proteins, particularly ATP-binding cassette (ABC) transporters, have been shown to play a vital role in facilitating transmembrane fatty acid movement, which regulates the spatial and temporal distribution of fatty acids and maintains the equilibrium between plant growth and defense requirements [18,19]. Among the ABC transporters, the ABCG subfamily is the most functionally diverse. This subfamily includes transporters involved in transporting signaling molecules, defensive metabolites, and hormones across membranes [20,21,22]. Research has shown that ABCG transporters play a critical role in disease response. In *Arabidopsis thaliana*, AtPDR12/AtABCG40 enables the transport of abscisic acid (ABA), thereby regulating stomatal closure and limiting pathogen entry [23,24]. PEN3/AtABCG36/AtPDR8 in *Arabidopsis thaliana* facilitates the efflux of defensive metabolites, increases callose deposition, and strengthens the physical barriers against pathogen invasion [25]. In *Nicotiana benthamiana*, NbABCG1 and NbABCG2 secrete antifungal diterpenoids while limiting the biosynthesis of pathogen-supportive metabolites, such as eugenol, during *Phytophthora infestans* infection [26]. In rice (*Oryza sativa*), OsABCG31 mediates the movement of resistance-related metabolites, thereby protecting *Magnaporthe oryzae* and *Rhizoctonia solani* [27]. In wheat (*Triticum aestivum*), Lr34 offers broad-spectrum resistance to fungal pathogens by transporting sinapyl alcohol, a precursor of cell wall lignification [28]. These findings demonstrate the functional diversity of ABCG transporters and their critical roles in plant disease resistance through the transport of metabolites and hormones.

Despite these significant studies on the roles of ABCG transporters in plant disease resistance, their specific functions in cotton, particularly in response to Verticillium wilt and Fusarium wilt, have not been studied thoroughly. Recent studies have identified stunted arbuscule (STR) and STR2 as fatty acid ABCG transporters in *Medicago truncatula*. These transporters facilitate the transport of plastid-synthesized fatty acids to the extracellular space and provide carbon sources for arbuscular mycorrhizal fungi [29]. Additionally, AP2/ERF, the key transcription factor in *Medicago*, is shown to regulate fatty acid transport by binding to the AW-box cis-regulatory element in the STR/STR2 promoter and activating its expression [30]. Based on these studies, this study aimed to understand the role of the homologous *GhSTR1* gene in Cotton. Using the amino acid sequence of *Medicago STR1* as a reference, the homologous *GhSTR1* gene was identified in cotton via sequence similarity analysis using BlastP. The expression of *GhSTR1* was found to be induced upon infection with *Verticillium dahliae* and *Fusarium oxysporum*. Functional analysis using virus-induced gene silencing (VIGS) indicated that *GhSTR1* is a negative regulator of disease resistance in cotton. To further validate its role, the homologous *AtSTR1* gene in *Arabidopsis* was deleted using T-DNA insertional mutagenesis (*Atstr1*). Loss of *AtSTR1* significantly increased the resistance to both Verticillium wilt and Fusarium wilt, further validating our findings in cotton. These findings indicate the functional role of *GhSTR1* as a negative regulator of cotton defense against Verticillium wilt and Fusarium wilt. Understanding *GhSTR1’s* role broadens our knowledge of ABCG transporters in plant–pathogen interactions in cotton and provides a novel approach to enhance disease resistance in cotton.

## 2. Results

### 2.1. Cloning and Bioinformatics Analysis of GhSTR1

Based on the *MtSTRI* function in interaction with arbuscular fungi, as indicated by many studies, a homologous protein in *Gossypium hirsutum* (cotton), designated as Gohir.A12G270100, was identified through homology searches in the Phytozome database and named *GhSTR1*. Using cDNA from the leaves of the cotton cultivar Junmian 1 as a template, the *GhSTR1* coding sequence (CDS) was successfully amplified via PCR, yielding a fragment of 2454 bp (Figure 1a). Bioinformatic analysis revealed that *GhSTR1* encodes a protein comprising 817 amino acids, predicted to be basic, hydrophilic, and unstable, with subcellular localization at the cell membrane. Using a homology-based approach, AtSTR1, a protein from Arabidopsis thaliana, was selected as a reference. Structural predictions using the SMART7 platform indicated that GhSTR1, MtSTR1, and AtSTR1 possess conserved ATPases associated with a variety of cellular activities and transmembrane helices, which are key characteristics of the ABCG subfamily of ABC transporters (Figure 1b). According to the HUGO (Human Genome Organization) nomenclature, ABC transporters are classified into eight subfamilies (ABCA–ABCH), with the ABCH subfamily absent in plants [21]. To examine the evolutionary relationship of GhSTR1, sequences from other plant ABC transporter gene families, including *Carya illinoinensis* (pecan), *Juglans regia* (walnut), *Alnus glutinosa* (alder), *Theobroma cacao* (cacao), *Citrus* x *clementina* (clementine), *Prunus avium* (cherry), and *Ricinus communis* (castor bean), were retrieved from GenBank. A phylogenetic tree was constructed using MEGA11 that revealed that GhSTR1 shares the closest evolutionary relationship with MtSTR1 and AtSTR1, with sequence similarities of 70.14% and 63.27%, respectively (Figure 1c). By integrating phylogenetic analysis and protein structural predictions, GhSTR1 was classified as a member of the ABCG subfamily, which is consistent with the classification of MtSTR1 and AtSTR1.

### 2.2. Expression Analysis of GhSTR1 Under V991 and St89 Stress

To investigate the expression pattern of the *GhSTR1* gene in cotton under stress from *V. dahliae* V991 (Verticillium wilt) and *F. oxysporum* St89 (Fusarium wilt), 15-day-old seedlings were inoculated using the root-dipping method [31]. Samples were collected from roots and true leaves at 0, 12, 24, 48, 72, 96, and 120 h post-inoculation (hpi). Water-treated plants (CK group) served as controls for analyzing the spatiotemporal expression characteristics of *GhSTR1* under these stress conditions. Under *V. dahliae* stress, *GhSTR1* expression in the leaves and roots showed dynamic variability. In leaves, the expression was significantly downregulated at 24 h and 96 h, followed by significant upregulation at 120 h (*p* < 0.05) (Figure 2a). In the roots, the expression followed a “rise–fall–rise–fall” pattern, with significant upregulation at 72 h and downregulation at 120 h (*p* < 0.05). No significant changes were observed at other time points (Figure 2b). Under *F. oxysporum* stress, a similar fluctuating expression pattern of *GhSTR1* was observed in both the leaves and roots. In leaves, significant downregulation occurred at 48 h, followed by considerable upregulation at 72 h and 120 h (*p* < 0.05) (Figure 2c). In the roots, expression was significantly upregulated at 96 h and downregulated at 120 h (*p* < 0.05), with no significant differences at other time points (Figure 2d). In the uninfected control cotton tissues (roots, stems, and true leaves), *GhSTR1* exhibited distinct tissue-specific expression patterns (Appendix A). The expression levels in stems were significantly higher than those in roots and leaves, suggesting a potential role in stem-specific physiological processes. These findings revealed that *GhSTR1* shows a notable spatiotemporal and tissue-specific expression pattern when subjected to *V. dahliae* and *F. oxysporum* stress, suggesting its potential involvement in the dynamic modulation of disease resistance mechanisms in cotton.

### 2.3. Construction of the GhSTR1 VIGS Vector and Verification of Silencing Efficiency

A VIGS targeting *GhSTR1* was successfully constructed where a 416 bp target fragment of *GhSTR1* was amplified by PCR and cloned into the TRV vector (Figure 3a). To confirm the cloning of the fragment, the TRV vector was digested with *Eco*RI and *Kpn*I, which yielded fragments of the expected sizes (Figure 3b). To confirm the accuracy of the TRV vector insert, we sequenced the cloned 416 bp fragment of *GhSTR1*. The sequencing data verified that the inserted fragment matches the *GhSTR1* target sequence without errors.

Cotyledons from cotton plants were infiltrated with the resuspended VIGS vector solution, and inoculated plants were kept in a growth chamber for 15 days at the conditions outlined in the methods. In the positive control plants carrying pTRV2::*GhCLA1*, a bleaching phenotype was observed (Figure 3c), demonstrating effective gene silencing. The expression levels of *GhCLA1* and *GhSTR1* were quantified by qRT-PCR. *GhCLA1* expression was significantly reduced in the leaves of pTRV2:: *GhCLA1* plants compared to pTRV2::*00* control, while *GhSTR1* expression was markedly downregulated in both roots and leaves of pTRV2::*GhSTR1* plants (Figure 3d,e). In summary, the TRV-based VIGS system effectively silenced *GhSTR1*, thus providing a reliable tool for its functional analysis.

### 2.4. Knockdown of GhSTR1 Enhances Cotton Resistance to V. dahliae and F. oxysporum

The role of *GhSTR1* in cotton resistance against *V. dahliae* (V991) and *F. oxysporum* (St89) was further investigated by using *GhSTR1*-silenced plants (pTRV2::*GhSTR1*), negative control plants (pTRV2::*00*), and wild-type (WT) plants. Following pathogen inoculation, phenotypic observations, disease index measurements, vascular browning assessments, and fungal biomass quantification were performed over a 20-day infection period. Silencing *GhSTR1* significantly enhanced cotton resistance to both pathogens. Compared to pTRV2::*00* and WT plants, the pTRV2::*GhSTR1* plants exhibited reduced symptoms of leaf chlorosis, wilting, and desiccation (Figure 4a,f). In particular, vascular browning in stems, which serves as an indicator of pathogen invasion severity, was markedly less severe in pTRV2::*GhSTR1* plants (Figure 4b,g). Disease index analysis showed a 50% reduction in disease severity in pTRV2::*GhSTR1* plants compared to that in controls (Figure 4c,h). Fungal biomass quantification confirmed these results, with fungal accumulation in pTRV2::*GhSTR1* plants reduced by over twofold compared to control plants for both *V. dahliae* and *F. oxysporum* (Figure 4d,i). Culturing of pathogens from stems further confirmed that fungal hyphal growth was significantly inhibited in pTRV2::*GhSTR1* plants compared to that in controls (Figure 4e,j). In summary, silencing *GhSTR1* significantly enhanced cotton resistance to *V. dahliae* and *F. oxysporum*, suggesting that *GhSTR1* functions as a negative regulator in disease resistance pathways.

### 2.5. Identification and Expression Analysis of the AtSTR1 T-DNA Insertion Homozygous Mutant

Based on the information in the database (http://signal.salk.edu/tdnaprimers.2.html) (accessed on 12 June 2024), the *AtSTR1* gene, located on chromosome 3 of *Arabidopsis thaliana*, consists of a single exon, with the T-DNA insertion site in the exon region of the SALK_129014 mutant (Figure 5a). Genotyping was conducted using SALK_129014-LP, SALK_129014-RP, and the T-DNA border primer, LBa1. PCR analysis indicated that the wild-type plants (Col-0) amplified a 1170 bp fragment with LP/RP primers, but no product was amplified with LBa1/RP primers. The homozygous mutant produced a 523–823 bp T-DNA fragment with LBa1/RP primers but no LP/RP product. The heterozygous mutant exhibited both bands. Plants 1 through 8 exclusively amplified the T-DNA fragment, confirming their homozygous status for *AtSTR1* (Figure 5b). To verify the loss of *AtSTR1* expression, the transcription levels in plant 1 (*Atstr1*) and wild-type plants were analyzed using SqRT-PCR and qRT-PCR. Both methods confirmed a significant reduction in *AtSTR1* expression in the mutant, with stable expression of the reference gene *Actin2* (Figure 5c,d). In conclusion, plant 1 (*Atstr1*) was characterized as a homozygous T-DNA insertion mutant, thus providing a robust model for investigating *AtSTR1* function.

### 2.6. Atstr1 Mutant Enhances Resistance to V. dahliae (V991) and F. oxysporum (St89) in Arabidopsis thaliana

To evaluate the role of *STR1* genes in disease resistance, the *Atstr1* mutant of *Arabidopsis thaliana* was tested against *V. dahliae* (V991) and *F. oxysporum* (St89). Following 15 days of infection, the *Atstr1* mutant showed significantly enhanced resistance compared to wild-type (Col-0) plants. The mutant showed decreased chlorosis, wilting, and growth inhibition (Figure 6a,e). Disease index analysis also confirmed a substantial reduction in disease severity within the mutant (Figure 6c,g). Stem cross-sections revealed moderate vascular browning in the *Atstr1* mutant compared to the severe browning observed in wild-type plants (Figure 6b,f). qRT-PCR analysis further attested to these findings by determining significantly lower fungal biomass in mutant than wild-type plants (Figure 6d,h). These findings indicate that the *AtSTR1* gene negatively influences disease resistance in *Arabidopsis*, implying a potentially conserved role of *STR1* genes in plant defense mechanisms. This research offers significant contributions to our understanding of the regulatory pathways underlying plant–pathogen interactions.

### 2.7. AtSTR1 Mutant Enhances Growth and Development in Arabidopsis thaliana

Plants with increased stress resistance often exhibit reduced growth, posing a challenge for resistance breeding [32]. To evaluate the effects of the *AtSTR1* mutation (*Atstr1*) on the growth and development of *Arabidopsis thaliana*, we recorded key morphological characteristics during the early growth phase (15 days) and reproductive stage (45 days). Phenotypic analysis indicated that the *Atstr1* mutant showed improved growth compared to wild-type (Col-0) plants (Figure 7a). The mutant demonstrated increased root elongation (Figure 7b), expanded rosette leaf size (Figure 7c,d), and a greater number of rosette leaves (Figure 7e) at the 15-day mark. By day 45, the *Atstr1* mutant showed accelerated bolting (Figure 7f), an overall great plant height (Figure 7g), and an increased number of bolting branches and siliques relative to the wild-type specimens (Figure 7h,i). These observations suggest that the absence of *AtSTR1* considerably enhanced both the vegetative and reproductive developmental traits. In conclusion, the functional loss of *AtSTR1* promoted growth vigor and development efficiency in *Arabidopsis*, highlighting its key role as a negative regulator of plant growth. These findings suggest the potential of targeting *AtSTR1* for improving growth traits in breeding programs.

### 2.8. Phenotypic Analysis of Atstr1 Mutant Under Drought Stress

To assess the effect of *AtSTR1* deletion on drought tolerance, wild-type (Col-0) and *Atstr1* mutant plants were grown in nutrient-rich soil for 25 days and then subjected to drought stress by withholding water. After 10 days of drought treatment, both genotypes showed typical drought symptoms, including leaf wilting, desiccation, curling, and slow growth, with no significant phenotypic differences observed between *Atstr1* mutant and wild-type plants (Figure 8a). Plants that were rated for survival following 8 days of rehydration also showed no significant differences between the two genotypes (Figure 8b), indicating that *AtSTR1* deletion does not affect drought tolerance in *Arabidopsis*. Water loss rate analysis further confirmed these findings. Across the eight time points, no significant differences in water retention patterns were detected between *Atstr1* mutant and wild-type plants (Figure 8c). These results align with those of the survival rate analysis, demonstrating that the loss of *AtSTR1* has no significant impact on drought tolerance in *Arabidopsis*. These findings suggest that *AtSTR1* may not play a critical role in drought stress responses, although it does have a key involvement in other abiotic stress.

## 3. Discussion

Verticillium wilt and Fusarium wilt significantly affect cotton production in China and drastically impact crop yield and quality. While current management strategies have shown some effectiveness, their use is limited due to their economic costs and environmental impact. Therefore, it is crucial to develop resistance in cotton plants using molecular breeding.

Previous studies have revealed that pathogen virulence is influenced by the availability of sugar and fatty acids provided by the host plant. This provides a novel approach to enhance plant resistance by regulating the nutrient supply from the host to the pathogen, thus restricting the pathogen [33]. For example, researchers have significantly improved plant resistance and disease reduction in maize smut, rice sheath blight, and cotton Verticillium wilt by silencing SWEET sugar transporter genes [33,34,35]. Previous research has also shown that fatty acids are the primary carbon source transferred from host plants to arbuscular mycorrhizal fungi [16,36]. In *Medicago truncatula*, the *STR*/*STR2* genes regulate lipid transport and play a crucial role in the formation of mycorrhizae [29].

Based on these studies and information on the *STR1* gene associated with fatty acid transport proteins, we identified its homologous gene in *Gossypium hirsutum* cv. Junmian1 using BlastP. The gene homologous to *STR1* was amplified using PCR. Bioinformatics analysis revealed that *GhSTR1* encodes a protein consisting of 817 amino acids. We characterized the gene and found it to be basic, hydrophilic, and unstable, and predicted to be localized to the cell membrane. SMART and phylogenetic analyses indicated that *GhSTR1* contains ATPase domain (AAA) associated with various cellular activities and six transmembrane domains, classifying it in the same ABCG subfamily as *MtSTR1* and *AtSTR1 (*Figure 1).

Previous studies have demonstrated that ABCG transporters are essential for plant resistance. Several examples from across different plant species have provided evidence to support this claim. In Arabidopsis thaliana, AtPDR12 [23] and AtABCG36 [24] have been shown to play a significant role in plant immunity. Similarly, NbABCG1 and NbABCG2 in tobacco [26], OsABCG31 in rice [27], and Lr34 in wheat [28] have been shown to regulate plant defense. These transporters move metabolites or signaling molecules, thereby controlling the response of plants to pathogens. However, the role of these transporters has not been explored deeply in cotton.

Given that these transporters are less studied in cotton, we investigated the expression patterns of *GhSTR1* in *Gossypium hirsutum* cv Junmian1 in response to *Verticillium dahliae* V991 and *Fusarium oxysporum* St89 inoculation. Our results from the study revealed that *GhSTR1* expression under pathogen-induced stress exhibited notable tissue specificity and temporal variations, following either “rise–fall–rise–fall” or “fall–rise–fall” patterns (Figure 2). These fluctuating gene expressions suggest that the plants use a multistage regulatory approach to combat pathogen invasion. The initial decrease in expression suggests that the plant restricts fatty acid transport to the pathogen to limit its growth. Conversely, the increase in gene expression indicates that the plant might be regulating its metabolic balance to enhance its defense. Furthermore, the tissue-specific expression observed in roots and leaves might be associated with infection routes and localized defense needs, implying that *GhSTR1* contributes to localized and systemic defense responses.

Recently, virus-induced gene silencing (VIGS) technology has been widely used to study gene function in cotton, owing to its high efficiency [37]. VIGS has been used to demonstrate the role of *GhPLP2* in regulating fatty acid metabolism and the Jasmonic acid signaling pathway, as well as its function in regulating resistance to Verticillium wilt [38]. In this study, we used VIGS to understand the role of the fatty acid transporter gene *GhSTR1* in cotton resistance to Fusarium wilt and Verticillium wilt. The results from our study indicated that silencing the expression of *GhSTR1* significantly increased resistance to both Fusarium wilt and Verticillium wilt. Plants with silenced *GhSTR1* showed reduced disease indices, reduced vascular browning, and decreased fungal biomass, suggesting enhanced resistance (Figure 4).

Although the negative regulatory role of *GhSTR1* has been studied using VIGS technology, the results may be limited by the instability and off-target effects inherent to VIGS. To further validate the function of *GhSTR1*, the role of *AtSTR1* (the homologous gene in *Arabidopsis thaliana*) was studied using T-DNA insertion mutagenesis. Under pathogen stress, the *Atstr1* mutant showed significantly lower disease indices, a reduction in vascular browning, and decreased fungal biomass compared to the wild-type (Col-0), indicating an increase in disease resistance (Figure 6). These findings are consistent with the functional identification of *GhSTR1* using VIGS and further confirm the role of *GhSTR1* in cotton’s response to *Verticillium dahliae* and *Fusarium oxysporum*. Interestingly, GhSTR1 is predicted to localize to the cell membrane, whereas its homolog AtSTR1 (also known as ABCG19) has been reported to localize to the vacuole membrane in *Arabidopsis thaliana* [39]. This discrepancy suggests that *STR1* genes may have context-dependent localization or dual roles in fatty acid transport. In the future, we should investigate whether GhSTR1 exhibits dynamic membrane localization under pathogen stress and whether this affects its role in disease resistance.

This research highlights the critical role of resistance genes in enhancing crop resistance to pathogens. However, practical breeding efforts require a comprehensive study of traits such as growth and development, drought tolerance, quality, and yield to ensure stable performance under multiple stress conditions [32]. This study showed that the *Atstr1* mutant showed overall better growth, including leaf area, root area, root length, and plant height, than the wild type (Figure 7). However, in the experiment on drought stress, there were no significant differences between mutant and wild-type plants in terms of survival rate and water loss (Figure 8). These results indicate a possible divergence in the functional roles of *AtSTR1* in disease resistance and drought tolerance.

Developing breeding techniques focused on nutrient delivery for disease resistance has been widely recognized as a practical and durable approach [40]. For example, CRISPR/Cas9 gene editing has been efficiently used to change TALE-binding elements, such as SWEET gene promoters, resulting in broad-spectrum resistance to bacterial blight in rice [41,42]. This research identified a novel modulator of cotton resistance to Fusarium and Verticillium wilts. This finding indicates the functional importance of ABCG transporters in plant–pathogen interactions. Our results suggest that *GhSTR1* is vital for cotton’s disease resistance mechanisms, although further research is required for deeper analysis of its function pathways.

Furthermore, *GhSTR1* may interact with other defense-related systems. Exploring these potential cross-regulatory interactions could significantly increase our understanding of cotton’s complex networks governing disease resistance. Moreover, gene-editing technologies could precisely control fatty acid transport between cotton and pathogenic fungi, an approach known as “starvation therapy.” Targeting the metabolic requirements of pathogens could effectively diminish the pathogen virulence and boost disease resistance in plants.

This study provides theoretical support for improving cotton disease resistance and suggests innovative gene editing and cross-pathway regulation strategies. To maximize their relevance in practical breeding programs, future studies should confirm these findings in multi-gene scenarios and under various environmental conditions.

## 4. Materials and Methods

### 4.1. Plant Materials and Growth Conditions

Seeds of *Gossypium hirsutum* cv. “Junmian 1” were surface-sterilized with 75% ethanol and rinsed with sterile water twice. They were soaked in 30% hydrogen peroxide (H_2_O_2_) at 28 °C for 3 h. Residual H_2_O_2_ was removed by rinsing the seeds 3 to 5 times with sterile conditions. Seeds were incubated in liquid Murashige and Skoog (MS) medium at 28 °C in the dark for 24–48 h until seed germination. Germinated seeds with uniform radicle lengths were subsequently transplanted to the matrix (the vermiculite and black soil in a volume ratio of 1:2). Sixty plants were used for this experiment. The pots were covered with plastic bags and kept under a 16 h light/8 h dark photoperiod at 28 °C with a light intensity of 120 μE·m^−2^·s^−1^ and relative humidity of 60–70%. The plastic bag was removed 5–7 days post-germination, and seedlings were left to grow in the above-mentioned conditions.

For *Arabidopsis thaliana*, Col-0 wild-type, and T-DNA insertion mutant, the seeds were surface-sterilized with 6% sodium hypochlorite for 5 min and rinsed 5 times with sterile water. The seeds were sown on 1/2 MS solid medium and placed at 4 °C in the dark for 3 days. The plates were kept in a growth chamber at 22 °C under a 16 h light/8 h dark photoperiod with a light intensity of 6000–8000 lx for 7 days. After seedling germination, the germinated plants were transplanted into a sterilized substrate of peat, vermiculite, and perlite (4:3:1, *v*/*v*/*v*) and grown at 23 °C in a growth chamber with 16 h light/8 h dark photoperiod.

### 4.2. Pathogen Growth and Inoculation

The fungal strains *Verticillium dahliae* V991 and *Fusarium oxysporum* St89 were used for plant inoculation. The strains were grown on potato dextrose agar (PDA) plates at 28 °C for 3–5 days. The fungal hyphae were then transferred to Czapek-Dox liquid medium (NaNO_3_, 0.3% *w*/*v*; KH_2_PO_4_, 0.1% *w*/*v*; MgSO_4_, 0.1% *w*/*v*; KCl, 0.1% *w*/*v*; FeSO_4_, 0.0002% *w*/*v*; and sucrose, 3% *w*/*v*; pH 6.0) and incubated at 28 °C with shaking at 180 rpm for 5–7 days.

Cotton plants were inoculated using the root dip method, as described by Zhang [31]. After 15–20 days of growth, the roots, stems, and fully expanded true leaves were sampled from some plants for tissue-specific expression analysis. At the same time, the seedlings with uniform root growth were selected, where roots were washed with water, and the seedlings were immersed in spore suspensions of *V. dahliae* V991 and *F. oxysporum* St89 at a concentration of 1 × 10^6^–10^7^ spores·mL^−1^ for 1 min. Seedlings treated with sterile water served as controls (CK). The roots and true leaves were sampled at 0, 12, 24, 48, 72, 96, and 120 h post-inoculation (hpi) for gene cloning and expression pattern analysis.

### 4.3. Analysis of GhSTR1 Gene Expression Patterns

Total RNA was extracted from collected leaves and roots sampled at 0, 4, 8, 12, 24, and 48 hpi using the Plant Polysaccharide and Polyphenol RNA Extraction Kit (Tiangen, Beijing, China), following the manufacturer’s protocol. First-strand cDNA was synthesized using the 5× All-In-One RT MasterMix kit (ABM, Zhenjiang, China). Quantitative RT-PCR (qRT-PCR) was performed on a LightCycler 480 system (Roche, Basilea Switzerland) using SYBR Premix Ex Taq (Takara, Beijing, China). Relative gene expression levels were calculated using the 2^−ΔΔCt^ method [43], with *GhUBQ7* (DQ116441) as the internal reference gene. Each experimental condition consisted of three biological replicates, each biological replicate including three technical replicates (*n* = 3 for each level). All the primers used for this analysis are listed in Appendix A.

### 4.4. Gene Cloning and Sequence Analysis

The amino acid sequence of *Medicago truncatula MtSTR1* (GenBank accession number: ACV73541.1) was used as a query to identify the homologous gene *GhSTR1* in upland cotton through a BlastP search in the Phytozome database (https://phytozome-next.jgi.doe.gov/) (12 June 2024). Primers specific to the *GhSTR1* sequence were designed using primer 5.0 (Appendix A). Amplification of the gene was conducted using Phusion High-Fidelity DNA Polymerase (NEB, Ipswich, MA, USA), following the manufacturer’s instructions. PCR amplicons were analyzed on 1% agarose gel electrophoresis, and the target bands were excised and cleaned using the Agarose Gel DNA Recovery Kit (Tiangen, Beijing, China). The purified amplicon fragments were ligated into the pEASY Blunt-Zero cloning vector and transformed into *Escherichia coli* DH5α competent cells (Tiangen, Beijing, China) according to manufacture protocol. The restriction enzyme digestion identified positive clones and sequenced by the Shanghai JieLi Biotechnology Co., Ltd. (Shanghai, China).

### 4.5. Sequence and Bioinformatics Analysis of GhSTR1

The open reading frame (ORF) of *GhSTR1* was analyzed using DNA Star17.6 software. Physicochemical properties, including the molecular weight, isoelectric point, instability index, and hydrophilicity, were predicted using the EXPASY ProtParam tool (https://web.expasy.org/protparam/) (7 May 2024). Subcellular localization was predicted using the LocTree tool (https://www.rostlab.org/services/loctree2/) (8 May 2024). Functional annotation and protein family classification were performed using Prosite (https://prosite.expasy.org/) (10 May 2024) and SMART tools (http://smart.embl-heidelberg.de/) (11 May 2024) [44]. Sequence alignment was conducted using Clustal X, and a phylogenetic tree was constructed using MEGA11 software [45]. These analyses provided insights into the evolutionary characteristics of *GhSTR1*.

### 4.6. Construction of Silencing Vectors and VIGS in Cotton

The Tobacco rattle virus (TRV)-mediated VIGS technology is well adapted to study the function of genes in cotton [46]. We used this technology to understand the function of *GhSTR1.* To suppress *GhSTR1* expression, the target sequences were designed using the SGN-VIGS online tool (https://vigs.solgenomics.net/) (12 June 2024). Primers were designed using DNAMAN software with *Eco*R1 and *Kpn*I restriction sites (Appendix A). The PCR product was digested with *Eco*R1 and *Kpn*I and then ligated into the pTRV2 vector using T4 DNA ligase. Ligated products were transformed into *E. coli* DH5α competent cells. Positive clones were confirmed by restriction digestion. Verified pTRV2::*GhSTR1* recombinant plasmids, along with pTRV:RNA1 and pTRV:RNA2 plasmids, were transformed into *Agrobacterium tumefaciens* GV3101 cells via electroporation. The *Agrobacterium* cells were injected into the cotyledons of 7-day-old plants using 1 mL syringes. The plants were then grown at 25 °C under 16 h /8 h light/dark cycle, as described by Li [5]. Approximately 15 days post-infection, bleaching of leaves was observed in the positive control group (pTRV2::*GhCLA1*). Root and true leaf samples were collected from the experimental group (pTRV2::*GhSTR1*) and control group (pTRV2::*00*). Silencing efficiency was assessed via qRT-PCR. Each experimental group included more than 60 seedlings, with three biological replicates per condition.

### 4.7. Genomic DNA Extraction and Homozygous Identification

DNA was extracted from fresh leaves of *Arabidopsis thaliana* seedlings using the EasyPure Plant Genomic DNA Extraction Kit (TransGen Biotech, Beijing, China). Gene-specific primers (LP and RP) and a T-DNA-specific primer (LBa1) were designed based on the T-DNA insertion site information for Salk129014 from the Salk Institute (http://signal.salk.edu/) (14 June 2024). PCR amplification was performed using genomic DNA extracted from Col-0 wild-type plants and mutant lines using the LP/RP and LBa1/RP primer combinations. The reaction conditions followed the EasyTaq DNA Polymerase protocol (TransGen Biotech, Beijing, China).

Total RNA was extracted from the leaves of homozygous *AtSTR1* mutant lines, and that of Col-0 wild-type plants was extracted using the EasyPure RNA Extraction Kit (TransGen Biotech, Beijing, China). RNA was reverse transcribed into cDNA using a Reverse Transcription Kit (TransGen Biotech, Beijing, China). Expression levels of *AtSTR1* were analyzed by qRT-PCR and semi-quantitative PCR (SqRT-PCR), using *Actin2* as the internal reference gene. Primer sequences are provided in Appendix A.

### 4.8. Disease Resistance Evaluation of Arabidopsis Atstr1 Mutant and Cotton VIGS Plants Against Fusarium and Verticillium Wilts

The severity of plant disease symptoms was categorized into four grades, ranging from 0 to 4 as follows. Grade 0: healthy plants with no visible symptoms. Grade 1: 0–25% of leaves show chlorosis or yellow spots. Grade 2: 25–50% of leaves exhibit yellow spots with slight leaf shedding. Grade 3: 50–75% of leaves display yellow or brown spots with moderate shedding. Grade 4: 75–100% of leaves are affected by yellow or brown spots, with significant leaf drop [47]. The Disease Index (DI) was calculated using the following formula: **DI = [(Disease Grade × Number of Infected Plants)/(Total Number of Plants × 4)] × 100**

Next, 3 mm stem pieces were excised 2 cm below the cotyledons from *pTRV::GhSTR1*-silenced and *pTRV::00* control plants after 15 days post-inoculation (dpi) with *Verticillium dahliae* or *Fusarium oxysporum*. These excised sections were sterilized with 75% sodium hypochlorite, rinsed with sterile water, and cultured on PDA medium containing 400 mg/L cefotaxime at 25 °C for 4 days to observe fungal regrowth [5]. DNA extraction was performed using the Plant Genomic DNA Extraction Kit (TransGen Biotech, Beijing, China), and qRT-PCR followed the protocol described by Luchi [48]. Fungal DNA’s ITS region was amplified with primers ITS-F and VE1-R, while internal reference genes were used (*UBQ7* for cotton and *Actin2* for *Arabidopsis*). This experiment was independently repeated with three technical and three biological replicates each time.

### 4.9. Growth and Development Phenotype Observation and Drought Stress Experiment for Arabidopsis Atstr1 Mutant

Wild-type (*Col-0*) and *Atstr1* mutant seedlings were transplanted into nutrient soil (vermiculite and black soil 1:2, *v*/*v*) for phenotype analysis. After 15 days, the plants were observed to determine overall plant morphology, root growth, rosette leaf diameter, and the number of rosette leaves. At 45 days, during the reproductive stage, additional phenotypes were assessed, such as plant height, leaf size, root characteristics, number of bolting branches, and number of siliques. A minimum of 40 plants per group was evaluated to ensure sufficient statistical power.

To assess drought tolerance, *Col-0* and *Atstr1* mutant plants were grown in nutrient-rich soil for 25 days under normal conditions. The plants were exposed to drought conditions by withholding irrigation for 10 days. Following re-watering, phenotypic changes were recorded, and the survival rates were calculated. For the water loss rate (WLR) measurements, 30-day-old plants were selected. The fresh weight (FW) of detached leaves was recorded initially and at 1 h intervals over a 15 h period.

### 4.10. Statistical Analysis

Results were presented as the mean ± standard deviation (SD) or standard error of the mean (SEM) based on three replicates. Statistical analyses were performed using GraphPad Prism 9.5 software (HM, San Diego, CA, USA). The significance levels were determined using *t*-tests or one-way analysis of variance (ANOVA). The levels of significance were indicated as follows: * *p* < 0.05, ** *p* < 0.01, and *** *p* < 0.001, representing varying degrees of statistical significance.

## 5. Conclusions

In this study, we determined the functional role of GhSTR1, a member of the ABCG subfamily of ATP-binding cassette (ABC) transporters that mediate cotton defense responses against *V. dahliae* and *F. oxysporum*. We identified *GhSTR1* as a homolog of *STR1* from *Medicago truncatula* and highlighted its evolutionary conservation and potential role in plant defense mechanisms. Expression profiling revealed that *GhSTR1* displays tissue-specific and spatiotemporal dynamics under stress conditions caused by *V. dahliae* and *F. oxysporum*. Functional validation was conducted using virus-induced gene silencing (VIGS), which showed that silencing *GhSTR1* enhanced disease resistance, as indicated by reduced symptom severity, vascular browning, and fungal biomass. Furthermore, *AtSTR1* loss-of-function mutants in *Arabidopsis thaliana* exhibit similar resistance phenotypes, highlighting the conserved regulatory role of *STR1* in pathogen defense. In addition to its role in disease resistance, the mutation of *AtSTR1* in *Arabidopsis* enhances the vegetative and reproductive growth of the plant, including increased root length, rosette leaf number, and plant height without compromising drought tolerance. These findings suggest that *GhSTR1* mediates a trade-off between defense and growth, offering a potential target for optimizing both traits for crop improvement.

## Figures and Tables

**Figure 1 plants-14-00465-f001:**
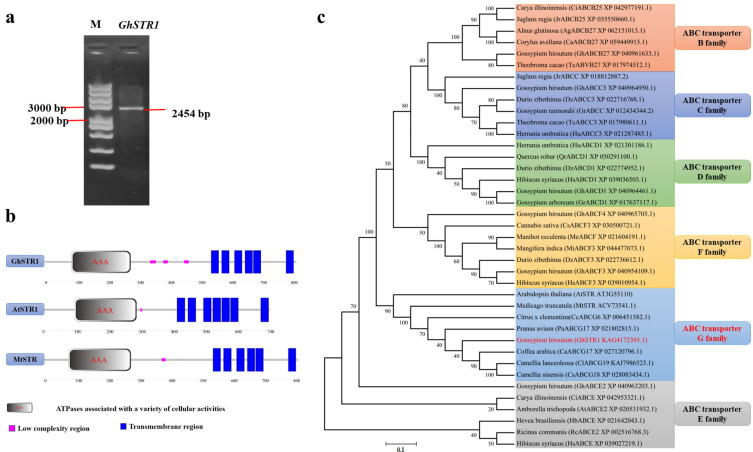
Cloning, structural analysis, and phylogenetic relationships of *GhSTR1.* (**a**) PCR amplification of the *GhSTR1* gene. The red arrow indicates the target band at the expected size of 2454 bp, confirming the successful cloning of the *GhSTR1* coding sequence (CDS). (**b**) Protein domain comparison. SMART-based domain predictions showed that *GhSTR1*, *MtSTR1*, and *AtSTR1* share a conserved AAA ATPase domain (red oval) and transmembrane helices (blue rectangles), which are characteristic features of the ABCG subfamily. (**c**) The phylogenetic analysis of GhSTR1 was conducted using MEGA11 to study the primary ABC transporter proteins from *Carya illinoinensis* (pecan), *Juglans regia* (walnut), *Alnus glutinosa* (alder), *Theobroma cacao* (cacao), *Citrus x clementina* (clementine*), Prunus avium* (cherry), and *Ricinus communis* (castor bean). The evolutionary relationships among these major ABC transporter proteins were analyzed using the Neighbor-Joining (NJ) method and the JTT substitution model in MEGA11 software (The red section of the figure illustrates the cotton proteins and their corresponding protein families analyzed in this study). Bootstrap analyses with 1000 replications were performed on the nodes of the phylogenetic tree to evaluate their statistical support. As shown in Figure 1c, the statistical support for key nodes confirms the robustness of the inferred evolutionary relationships. The phylogenetic tree indicates that GhSTR1 is closely related to MtSTR1 and AtSTR1, confirming its classification within the ABCG subfamily.

**Figure 2 plants-14-00465-f002:**
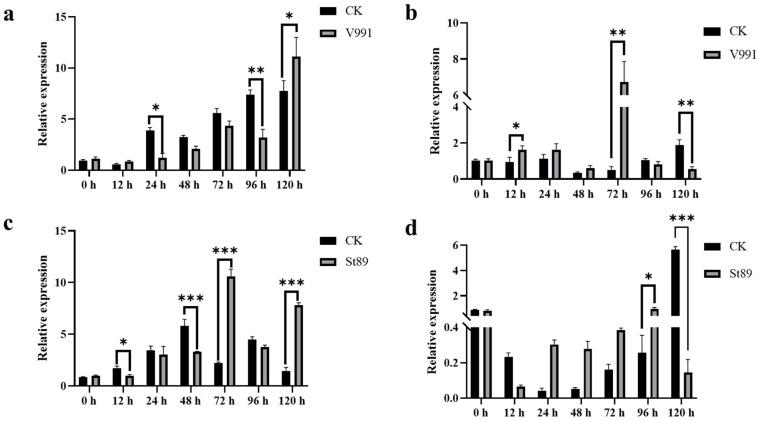
Transcript levels of *GhSTR1* under *Verticillium dahliae* V991 and *Fusarium oxysporum* St89 stress. (**a**,**c**) Relative expression levels of *GhSTR1* in leaves under stress from *V. dahliae* V991 and *F. oxysporum* St89, respectively. (**b**,**d**) Relative expression levels of *GhSTR1* in roots under stress from *V. dahliae* V991 and *F. oxysporum* St89, respectively. Data are expressed as the mean ± standard error (*n* = 3) and normalized to the control group (CK, sterile water treatment). Statistical analysis was conducted using the *t*-test, with significance indicated as follows: * *p* < 0.05, ** *p* < 0.01, and *** *p* < 0.001.

**Figure 3 plants-14-00465-f003:**
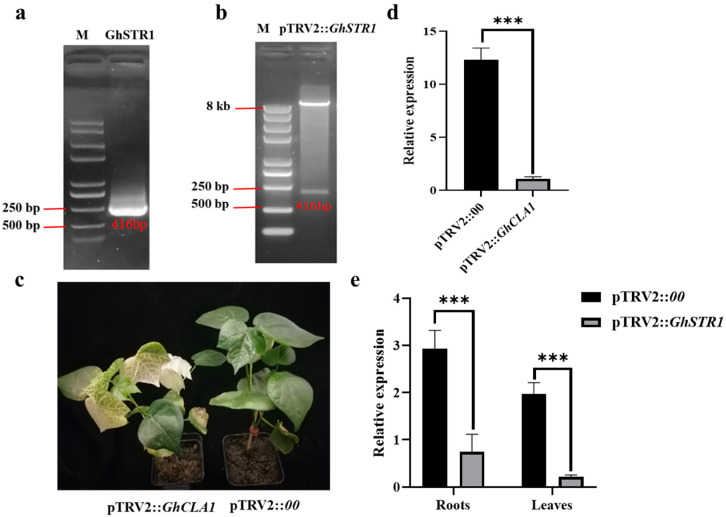
Silencing efficiency of the *GhSTR1* gene. (**a**) PCR amplification of the *GhSTR1* target fragment. (**b**) Restriction digestion of the TRV vector, confirming successful vector construction. (**c**) The bleaching phenotype observed in pTRV2::*GhCLA1*-silenced cotton plants, demonstrating effective gene silencing. (**d**,**e**) Relative expression levels of *GhCLA1* and *GhSTR1* in pTRV2::*00* and pTRV2::*GhSTR1* plants, respectively. Data are presented as the mean ± standard error (*n* = 3). Statistical significance is indicated as follows: *** *p* < 0.001.

**Figure 4 plants-14-00465-f004:**
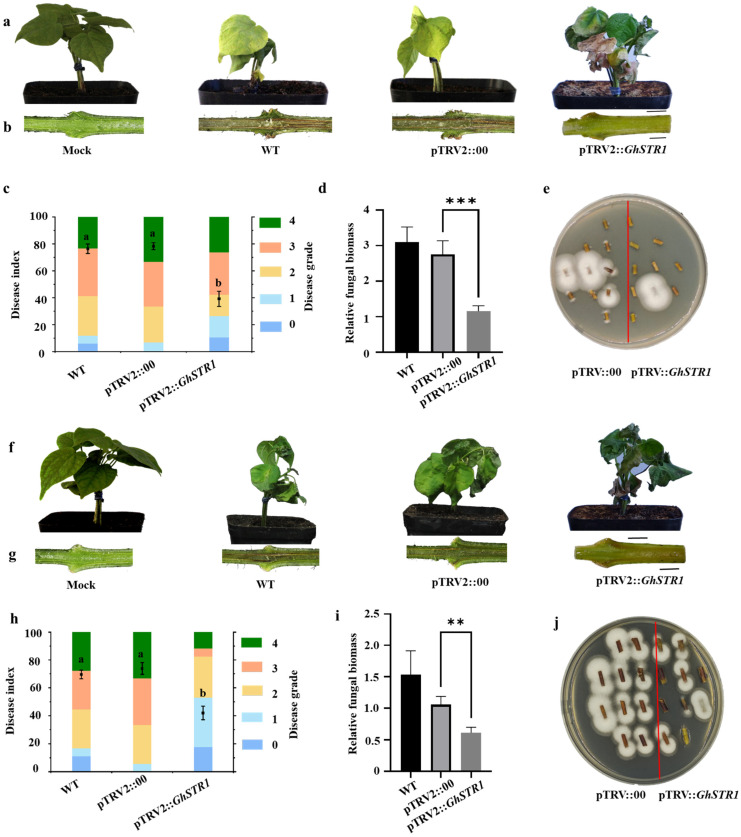
Effects of *GhSTR1* gene silencing in cotton resistance to *V. dahliae* V991 and *F. oxysporum* St89. (**a**,**f**) Leaves of pTRV2::*GhSTR1* plants exhibited more severe chlorosis, wilting, and lesions following infection with *V. dahliae* (V991) and *F. oxysporum* (St89) compared to WT and pTRV2::*00* controls, respectively. Scale bar = 2 cm. (**b**,**g**) Longitudinal sections of infected stems showed more pronounced vascular browning in pTRV2::*GhSTR1* plants, indicating greater pathogen invasion. Scale bar = 0.2 cm. (**c**,**h**) Disease index analysis at 20 dpi revealed significantly higher indices in pTRV2::*GhSTR1* plants than WT and pTRV2::*00* control. (**d**,**i**) qRT-PCR analysis showed significantly higher fungal biomass in pTRV2::*GhSTR1* plants than in the controls. (**e**,**j**) Fungal hyphal growth in stem sections (1 cm above the cotyledonary node) cultured on PDA medium was significantly greater in pTRV2::*GhSTR1* plants than in the controls. Scale bar = 0.2 cm. Each group included ≥30 plants with 3 replicates to ensure result reliability. Data are expressed as the mean ± standard error (*n* = 3). Statistical significance was assessed using analysis of variance (ANOVA), followed by Duncan’s multiple comparison test. The significance levels are indicated as follows: ** *p* < 0.01, and *** *p* < 0.001. Different groups with different letters represent statistically significant differences (*p* < 0.05).

**Figure 5 plants-14-00465-f005:**
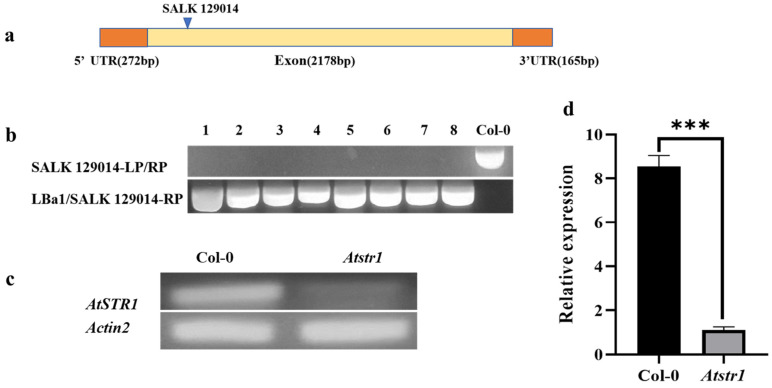
Genotypic validation and expression analysis of *AtSTR1* T-DNA insertion mutant in *Arabidopsis thaliana.* (**a**) Schematic representation of *AtSTR1* gene structure in the SALK_129014 mutant. The promoter is shown as an orange rectangle, the single exon as a yellow rectangle, and the T-DNA insertion site as a blue inverted triangle. (**b**) Genotyping results for the homozygous SALK_129014 mutant. Homozygous plants lacked amplification with LP/RP primers but showed a T-DNA-specific fragment with LBa1/RP primers. (**c**) SqRT-PCR showed reduced *AtSTR1* expression in the *Atstr1* mutant compared to that in the wild-type plants. *Actin2* was used as the reference gene for normalization. (**d**) qRT-PCR confirmed significantly reduced *AtSTR1* expression in the *Atstr1* mutant relative to the wild-type plants. Data are presented as mean ± standard error (*n* = 3), with statistical significance indicated as follows: *** *p* < 0.001.

**Figure 6 plants-14-00465-f006:**
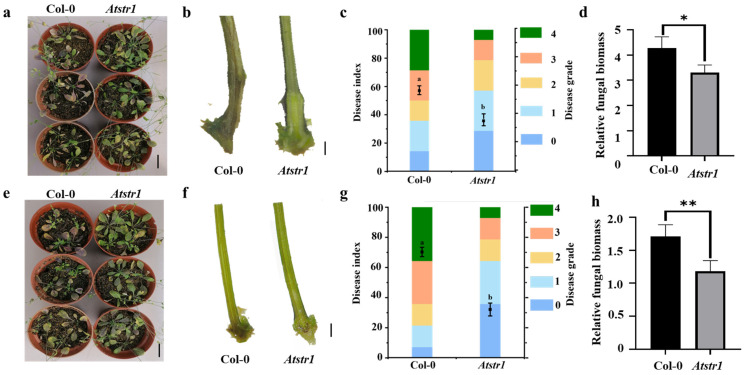
Enhanced resistance of *Atstr1* mutant to *V. dahliae* (V991) and *F. oxysporum* (St89). (**a**,**e**) Phenotypic comparison of *Arabidopsis thaliana* Col-0 wild-type and *Atstr1* mutant 15 days post-infection (dpi) with *V. dahliae* (V991) and *F. oxysporum* (St89), respectively. *Atstr1* mutant displayed reduced wilting and chlorosis compared to the wild-type plants. Scale bar = 1 cm. (**b**,**f**) Stem longitudinal sections 1 cm above the tillering node, showing vascular browning at 15 dpi with V991 and St89. *Atstr1* mutant exhibited milder vascular browning compared to the wild-type plants. Scale bar = 0.2 cm. (**c**,**g**) Disease index values at 15 dpi. *Atstr1* mutant showed significantly lower disease indices than the wild-type plants for both V991 and St89 infections. (**d**,**h**) qRT-PCR analysis of the fungal biomass at 15 dpi. *Atstr1* mutant exhibited significantly reduced fungal biomass compared to wild-type plants. Data are expressed as mean ± standard error (*n* = 3). Statistical significance was assessed using analysis of variance (ANOVA), followed by Duncan’s multiple comparison test. Significance levels are indicated as follows: * *p* < 0.05, ** *p* < 0.01. Groups with different letters represent statistically significant differences at *p* < 0.05.

**Figure 7 plants-14-00465-f007:**
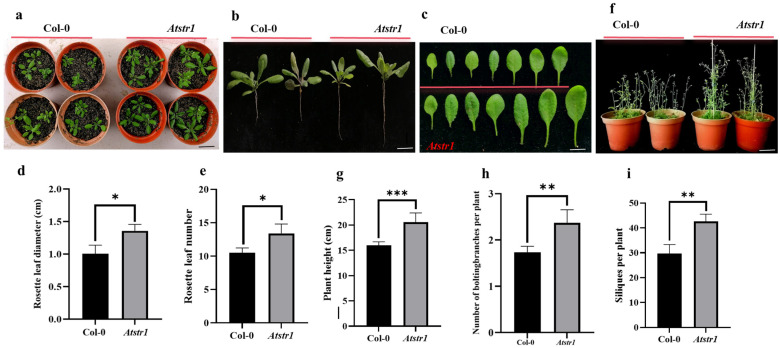
Growth and development phenotype analysis of *Atstr1* mutant in *Arabidopsis thaliana.* (**a**) Overall developmental state of the wild-type (Col-0) and *Atstr1* mutant during the 15-day growth stage. Scale bar = 5 cm. (**b**) Root length measurements during the 15-day growth period. Scale bar = 1 cm. (**c**) Leaf size of the wild-type and *Atstr1* mutant during the 15-day growth stage. Scale bar = 2 mm. (**d**) Rosette leaf diameter during the 15-day growth stage. The average diameter was measured at the widest point of the leaf blade across all rosette leaves of the plant. (**e**) Rosette leaf number during the 15-day growth stage. (**f**) Overall developmental state of the wild-type and *Atstr1* mutant during the 45-day growth stage. Scale bar = 5 cm. (**g**) Plant height during the 45-day growth stage. (**h**) Number of bolted branches per plant during the 45-day growth stage. (**i**) Number of siliques per plant during the 45-day growth stage. Forty plants were analyzed for each treatment. Data are presented as the mean ± standard error (SEM; *n* = 3). The statistical significance is as follows: * *p* < 0.05, ** *p* < 0.01, and *** *p* < 0.001.

**Figure 8 plants-14-00465-f008:**
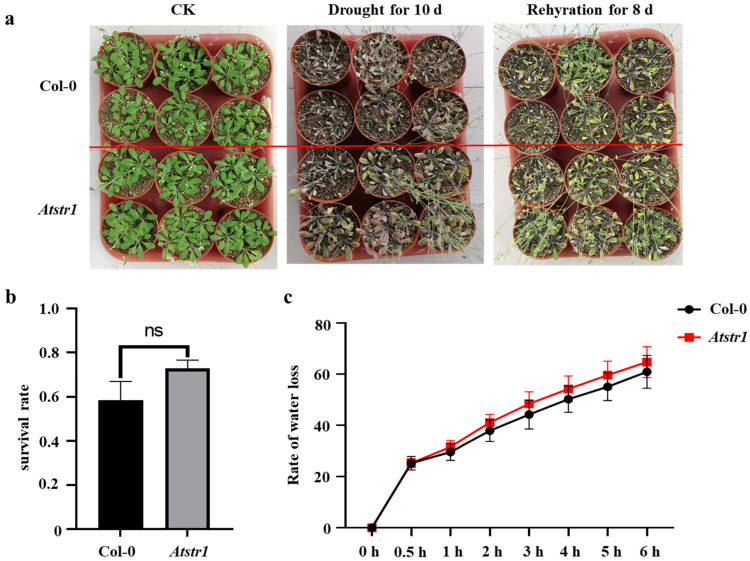
Phenotypic and physiological analysis of *Atstr1* mutant under drought stress. (**a**) Phenotypic comparison of *Arabidopsis thaliana* Col-0 wild-type and *Atstr1* mutant plants before drought treatment, after 10 days of drought stress, and following 8 days of rehydration. (**b**) Survival rate analysis of Col-0 and *Atstr1* mutant plants after drought stress and rehydration. (**c**) Water loss rate curves comparing Col-0 and *Atstr1* mutant plants during drought stress. A total of 40 plants were analyzed per treatment. Data are presented as mean ± standard error (SEM; *n* = 3). Statistical significance is as follows: ns: no significant difference.

## Data Availability

The data that support the findings of this study are available from the corresponding author upon reasonable request.

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
