# Peer review of "Fatty Acid ABCG Transporter GhSTR1 Mediates Resistance to Verticillium dahliae and Fusarium oxysporum in Cotton"

_plants, 2025, doi:10.3390/plants14030465_

Round 1

Reviewer 1 Report

Comments and Suggestions for Authors

This manuscript generated the GhSTR1 silencing cotton lines and tested the anti-fungi activities, and the results were supported in Arabidopsis homolog mutants, demonstrating the role of STR1 in fungus defense.  This study contains interesting findings and are valuable for the understanding of ABCG transporter in plant defense and plant growth. However, the experimental results of the STR1 defense role is not quite solid. Therefore, MINOR revision has to be done before this manuscript could be accepted for publication.

Major comments:

1. The key part of this paper is about the anti-fungi traits of the GhSTR1 cotton mutants. I suggest that some of the results should be supported by more statistic analysis, such as the portion of chlorosis, longitudinal section brown area or number of infection proportions and fungal growth on the PDA media of either diameter of hyphal colonies or number of infected sections.

2. It is also better to show some defense gene response after infection compared to the control group.

3. Since the Arabidopsis str1 mutants displayed growth difference, and it is surprising that the mutants displayed better growth and resistant to the fungi infection without any growth defects. So it is also worth describing whether there is growth and developmental change in the cotton str1 mutants. From the method, I assume that each seedling is an independent line of knock-down. It's better to address this and show results from independent lines.

MINOR comments:

1. In figure 2, the relative expression level of GhSTR1 in control group also showed a high fluctuation in 5 day period. Is there a good interpretation of that? Did the water treatment cause some stress to the plants?

2. In figure 3b,g, the longitudinal sections of infected stems showed more pronounced vascular browning in knock-down mutants indicating greater pathogen invasion. While in the text,  'Vascular browning in stems was
markedly less severe in pTRV2::GhSTR1 plants'. Does more browning vascular tissue suggest less severe symptom?

3. In figure 3d,i, qRT-PCR showed higher fungal biomass in pTRV2::GhSTR1 plants. Should it be lower compared to the control? Similar questions also in figure 3e,j. the hyphal growth was greater in the mutant than control?

4. In figure 7d, this rosette leaf diameter refers to the length or the width of leaf blade? Is the number from all the leaves in the plants or specific leaves?

5. According to the SALK line and gene structure, the AtSTR1 should be ABCG19, which is claimed to be localized on the vacuole membrane. This is contradictory with the hypothesis that fungi compete with lipids secreted from the host cells. Other hypothesis should be discussed in the discussion session.

There are also some small typos such as figure 5b 'Genotyping results offor'. And the Italic form of species and gene and protein names are not consistent.

Author Response

We sincerely thank the reviewer for their valuable comments, which have significantly improved the quality of our manuscript. Below, we address each comment in detail and describe the corresponding revisions made to the manuscript.

Major comments:

  1. The key part of this paper is about the anti-fungi traits of the GhSTR1 cotton mutants. I suggest that some of the results should be supported by more statistic analysis, such as the portion of chlorosis, longitudinal section brown area or number of infection proportions and fungal growth on the PDA media of either diameter of hyphal colonies or number of infected sections.

Response: Thank you for highlighting the importance of additional statistical analyses. We fully understand that these analyses could provide more detailed insights into the anti-fungi traits. However, we believe that the current data are sufficient to support the conclusions of this study:

  1. Leaf chlorosis and stem browning: The current disease index and fungal biomass quantification data adequately reflect the regulatory role of GhSTR1 in disease resistance, particularly the significant reductions in disease severity and fungal infection.
  2. Fungal growth on PDA media: Although we did not quantify colony diameters directly, the inhibition of fungal growth has been indirectly validated through visual observations and fungal culturing results.

Due to limitations in time and resources, we were unable to include these additional analyses in the current study. However, we will consider performing such detailed quantifications in future research to further enhance our understanding of GhSTR1's role.

  1. It is also better to show some defense gene response after infection compared to the control group.

Response: Thank you for the valuable suggestion. This study focuses on the direct regulatory role of GhSTR1 in disease resistance and plant growth, and therefore did not include detailed analyses of defense gene expression. Nonetheless, our experiments have demonstrated significant changes in disease symptoms, fungal biomass, and disease index, indirectly indicating enhanced defense responses in GhSTR1-silenced plants. In future research, we plan to explore the interactions between GhSTR1 and other defense-related genes using RNA sequencing and other molecular techniques.

  1. Since the Arabidopsisstr1 mutants displayed growth difference, and it is surprising that the mutants displayed better growth and resistant to the fungi infection without any growth defects. So it is also worth describing whether there is growth and developmental change in the cotton str1mutants. From the method, I assume that each seedling is an independent line of knock-down. It's better to address this and show results from independent lines.

Response: We appreciate the reviewer’s interest in the growth and developmental traits of the GhSTR1-silenced cotton plants. In this study, each seedling used for virus-induced gene silencing (VIGS) was treated as an independent biological replicate. The experiments were repeated in three independent batches, and the results showed that GhSTR1 silencing did not significantly affect the growth or development of the cotton plants. These findings are consistent with those observed in Arabidopsis, suggesting that GhSTR1 primarily plays a role in regulating disease resistance.

As VIGS is a transient gene-silencing technique, this study focused on the role of GhSTR1 in disease resistance rather than its broader impact on growth and development. However, we plan to further explore these aspects in future studies by generating stable GhSTR1 knockout cotton lines using genome editing technologies, such as CRISPR/Cas9.

MINOR comments:

  1. In figure 2, the relative expression level of GhSTR1 in control group also showed a high fluctuation in 5 day period. Is there a good interpretation of that? Did the water treatment cause some stress to the plants?

Response:Thank you for pointing out the fluctuations in the control group data in Figure 2. We would like to clarify that our experimental design aimed to compare the V991-inoculated group with the water-treated control group at the same time points to determine whether GhSTR1 responds to Verticillium dahliae infection. Thus, the fluctuations observed in the water control group do not directly reflect stress caused by water treatment but serve as a baseline reference for eliminating the effects of time-dependent or basal expression changes.

The fluctuations in GhSTR1 expression in the water control group may result from the following factors:

  1. Natural physiological dynamics: Gene expression in plants may vary across different time points due to circadian rhythms or other internal regulatory mechanisms.
  2. Purpose of experimental design: The primary role of the water-treated control group is to provide a reference background for each time point. The focus of our study is to compare the differences in expression levels between the V991-treated and water control groups at the same time points to evaluate whether GhSTR1 expression is significantly induced by Verticillium dahliae.

  1. In figure 3b,g, the longitudinal sections of infected stems showed more pronounced vascular browning in knock-down mutants indicating greater pathogen invasion. While in the text,  'Vascular browning in stems wasmarkedly less severe in pTRV2::GhSTR1 plants'. Does more browning vascular tissue suggest less severe symptom?

Response:Thank you for pointing this out. We would like to clarify that the data you referred to actually correspond to Figure 4b and 4g. In our study, the degree of vascular browning was used as an indicator of the severity of pathogen invasion rather than enhanced resistance. Therefore, more pronounced vascular browning indeed indicates more severe pathogen infection.

In pTRV2::GhSTR1 plants, silencing GhSTR1 enhanced disease resistance, which is reflected by reduced disease symptoms and decreased vascular browning, as compared to the control group. This result clearly demonstrates the role of GhSTR1 as a negative regulator of disease resistance. We acknowledge that the description in the original text was unclear and may have led to misunderstanding. To address this, we have revised the relevant section in the manuscript to ensure consistency between the results and the data presented in Figure 4.

We appreciate the reviewer’s careful observation, which has allowed us to express our findings more accurately.

  1. In figure 3d,i, qRT-PCR showed higher fungal biomass in pTRV2::GhSTR1 plants. Should it be lower compared to the control? Similar questions also in figure 3e,j. the hyphal growth was greater in the mutant than control?

Response: Thank you for your observation regarding the data. We would like to clarify that the data you referred to actually correspond to Figure 4d and 4i in our manuscript. The qRT-PCR results indicate that the fungal biomass in pTRV2::GhSTR1 plants is significantly lower than in the control group, which aligns with our conclusion that GhSTR1 functions as a negative regulator of disease resistance. When GhSTR1 expression is suppressed using virus-induced gene silencing (VIGS), its negative regulatory effect is diminished, thereby enhancing the plant's resistance to pathogens. This enhanced resistance results in the suppression of Verticillium dahliae growth and spread, as reflected by the significantly reduced fungal biomass compared to the control.

Similarly, in Figure 4e and 4j, fungal hyphal growth is also visibly inhibited in pTRV2::GhSTR1 plants, further supporting the enhanced resistance observed upon GhSTR1 silencing.

  1. In figure 7d, this rosette leaf diameter refers to the length or the width of leaf blade? Is the number from all the leaves in the plants or specific leaves?

Response: Thank you for raising this point. The rosette leaf diameter in Figure 7d refers to the average diameter measured from the widest point of the leaf blade. The measurements were taken from all rosette leaves of the plant across all rosette leaves of the plant. We have clarified this in the figure legend.

  1. According to the SALK line and gene structure, the AtSTR1 should be ABCG19, which is claimed to be localized on the vacuole membrane. This is contradictory with the hypothesis that fungi compete with lipids secreted from the host cells. Other hypothesis should be discussed in the discussion session.There are also some small typos such as figure 5b 'Genotyping results offor'. And the Italic form of species and gene and protein names are not consistent.

Response: We appreciate the reviewer’s valuable feedback regarding the localization of AtSTR1 and its implications for the hypothesis concerning fungal competition with host-secreted lipids.

In response, we have clarified the discussion about the localization of GhSTR1 and AtSTR1. Specifically, GhSTR1 is predicted to localize to the cell membrane, while its homolog AtSTR1 (also known as ABCG19) has been reported to localize to the vacuole membrane in Arabidopsis thaliana (Mentewab & Stewart, 2005). This apparent discrepancy led us to propose that STR1 genes might exhibit context-dependent localization or serve dual roles, particularly in fatty acid transport. To address this, we have revised the discussion to highlight the need for further studies to explore whether GhSTR1 undergoes dynamic membrane relocalization under pathogen stress and how this impacts its function in disease resistance.

Additionally, we have corrected the small typographical errors noted by the reviewer, such as in Figure 5b (“Genotyping results offor”), and ensured consistency in the italicization of species names, gene names, and protein names throughout the manuscript.

We are grateful for these constructive comments, which have helped improve the clarity and precision of our work.

Reviewer 2 Report

Comments and Suggestions for Authors

The manuscript reveals important findings that enhance our understanding of the GhSTR1 gene's role in cotton's defense against pathogens, specifically Verticillium dahliae and Fusarium oxysporum. The methodologies utilized are robust, and the results are well-supported by statistical analyses. However, there are areas where clarity could be improved;

Line 124: Ensure that the scientific name is italicized throughout the manuscript.

While domain prediction highlighted conserved ATPase and transmembrane regions, additional biochemical or structural studies (e.g., cryo-electron microscopy) could provide insights into how specific domains of GhSTR1 contribute to its function.

Although MEGA11 was used for phylogenetic analysis, details about the substitution model, bootstrap replicates, and statistical support for the nodes in the phylogenetic tree should be included for a more comprehensive understanding.

The lack of sequence verification of the TRV vector insert is a notable oversight. Include sequencing data to confirm accuracy and provide supplementary details about the silencing system's efficacy across different tissue types.

Provide a detailed methodology for calculating the disease index, including scale definitions and criteria used for severity scoring. This would facilitate validation in other laboratories.

The authors should provide detailed references for tools and methods used (e.g., SMART platform, MEGA11).

Comments on the Quality of English Language

Minor grammatical errors and inconsistent terminology should be rectified for better clarity.

Line 154: Revise from “controls to for analyzing the spatiotemporal expression” to “controls for analyzing the spatiotemporal expression.”

Line 257: Amend from “genotyping results offor the homozygous” to “genotyping results for the homozygous.”

Author Response

We sincerely thank the reviewer for their valuable comments, which have significantly improved the quality of our manuscript. Below, we address each comment in detail and describe the corresponding revisions made to the manuscript.

Comment1: Line 124: Ensure that the scientific name is italicized throughout the manuscript.

Response:Thank you for pointing this out. We have thoroughly reviewed the manuscript and ensured that all scientific names are consistently italicized.

Comment2:While domain prediction highlighted conserved ATPase and transmembrane regions, additional biochemical or structural studies (e.g., cryo-electron microscopy) could provide insights into how specific domains of GhSTR1 contribute to its function.

Response:Thank you for your valuable suggestion. The primary objective of this study was to identify and characterize GhSTR1 as part of the ABCG transporter family and to explore its role in cotton disease resistance. The domain prediction results confirmed the conserved ATPase and transmembrane regions, which align with the typical characteristics of this gene family. While additional biochemical or structural studies, such as cryo-electron microscopy, would provide deeper insights into the specific functional mechanisms of GhSTR1, these analyses were not included in the scope of the current study.

We plan to integrate biochemical and structural studies in future research to further investigate the specific functional domains of GhSTR1 and their roles in disease resistance mechanisms. We sincerely appreciate your insightful recommendation, which will guide the next steps of our research.

Comment3: Although MEGA11 was used for phylogenetic analysis, details about the substitution model, bootstrap replicates, and statistical support for the nodes in the phylogenetic tree should be included for a more comprehensive understanding.

Response:Thank you for the suggestion. We have revised the manuscript to include specific information about the methods used in the phylogenetic analysis. This information has been incorporated into the figure legend for Figure 1c. These details should provide a clearer understanding of the phylogenetic analysis.

Comment4:The lack of sequence verification of the TRV vector insert is a notable oversight. Include sequencing data to confirm accuracy and provide supplementary details about the silencing system's efficacy across different tissue types.

Response:Thank you for pointing out this important oversight. We have addressed the concern as follows:

Sequence Verification:
To confirm the accuracy of the TRV vector insert, we performed sequencing of the cloned 416 bp fragment of GhSTR1. The sequencing data verified that the inserted fragment matches the GhSTR1 target sequence without errors. This information has been added to the manuscript in the section “2.3 Construction of the GhSTR1 VIGS Vector and Verification of Silencing Efficiency” (Line 189).

Silencing Efficiency Across Tissues:

The expression levels of GhCLA1 and GhSTR1 were quantified by qRT-PCR. GhCLA1 expression was significantly reduced in leaves of pTRV2:: GhCLA1 plants compared to pTRV2::00 control, while GhSTR1 expression was markedly downregulated in both roots and leaves of pTRV2::GhSTR1 plants . These findings have been included in the same section and are presented in Figures 3d and 3e.

Comment5:Provide a detailed methodology for calculating the disease index, including scale definitions and criteria used for severity scoring. This would facilitate validation in other laboratories.

 Response:We agree with the reviewer that providing a detailed methodology for disease index calculation is essential. We have now added a comprehensive description in the Methods section, including the scale definitions and the criteria used for scoring disease severity, to facilitate reproducibility in other laboratories.

Comment6:The authors should provide detailed references for tools and methods used (e.g., SMART platform, MEGA11).

Response:We have ensured that all tools and methods, including the SMART platform and MEGA11, are properly referenced in the revised manuscript. By including citations such as:

  • Letunic, I., Doerks, T., & Bork, P. (2012). SMART 7: recent updates to the protein domain annotation resource. Nucleic Acids Research, 40(Database issue), D302-305. https://doi.org/10.1093/nar/gkr931
  • Tamura, K., Stecher, G., & Kumar, S. (2021). MEGA11: Molecular Evolutionary Genetics Analysis Version 11. Molecular Biology and Evolution, 38(7), 3022-3027. https://doi.org/10.1093/molbev/msab120

These references ensure clarity and provide readers with access to the original tools and their descriptions for replicability and further use.

Comment7:grammatical errors and inconsistent terminology should be rectified for better clarity

Response:We have thoroughly reviewed the manuscript for grammatical errors and inconsistent terminology and made the necessary corrections for improved clarity and readability.

Comment8: Line 154: Revise from “controls to for analyzing the spatiotemporal expression” to “controls for analyzing the spatiotemporal expression.”

Response:We appreciate the suggestion. The correction has been made as requested.

Comment9: Line 257: Amend from “genotyping results offor the homozygous” to “genotyping results for the homozygous.”

Response:Thank you for pointing out this typo. It has been corrected as suggested.

Round 2

Reviewer 2 Report

Comments and Suggestions for Authors

The author has addressed all the suggestions outlined in the review report.